# Short- and Long-Term Effects of an Intervention to Act against Sexual Violence in Sports

Alina Schäfer-Pels [1,*], Jeannine Ohlert [2], Thea Rau [1] and Marc Allroggen [1]

1 Department of Child and Adolescent Psychiatry/Psychotherapy, University Hospital Ulm, 89075 Ulm, Germany
2 Institute of Psychology, German Sport University Cologne, 50933 Cologne, Germany
* Correspondence: alina.schaefer-pels@uniklinik-ulm.de

**Abstract:** In recent years, an increasing number of cases of sexual violence (SV) in organized sports have received worldwide attention. To counteract the emergence of SV, various preventive measures have been developed and implemented. However, the effectiveness of these preventive measures has not been adequately tested. To close this gap, the purpose of the present study was to evaluate the effectiveness of a workshop intervention that was conducted within the context of organized sports in Germany. The one-day workshop intervention was conducted with 137 stakeholders in organized sports (coaches, athletes, board members, and parents). The intervention was evaluated by measuring the short-term (immediately before and after the workshop) and long-term effects (six months after the workshop). The analyses showed positive short-term (such as on attitudes toward SV and the intention to act against SV) and positive long-term effects (on knowledge about SV and a culture of prevention in the sports club and club behavior) of the workshop. The workshop was effective in the short term and the long term regarding the most relevant indicators (i.e., taking measures against SV). Therefore, it can be concluded that more workshops should be held in clubs in order to sensitize stakeholders and foster measures against SV in sports.

**Keywords:** sports; prevention; sexual violence; theory of planned behavior

## 1. Introduction

Sexual violence has now been acknowledged as an issue that needs to be addressed by sports organizations around the world. There are several proposed methods for preventing the occurrence of sexual violence in sports clubs and federations, including establishing rules for coaches, naming a contact person for child protection, and workshops to sensitize actors in organized sports to the topic. However, evaluations of the effectiveness of these preventive measures are scarce. The purpose of this study was thus to evaluate the effect of a sensitizing workshop intervention in organized sports in Germany on factors such as knowledge about sexual violence, personal attitudes toward the prevention of sexual violence, the intention to act against sexual violence, and preventive measures taken against sexual violence by the sports organization involved. The theory of planned behavior (Ajzen 1991) was used as a theoretical background.

Sexual violence in sports is commonly defined as any form of "behaviour towards an individual or group that involves sexualized verbal, nonverbal or physical behavior, whether intended or unintended, legal or illegal, that is based upon an abuse of power and trust and that is considered by the victim or a bystander to be unwanted or coerced" (IOC Medical Commission Expert Panel 2007, p. 3). In a recent study, almost 31% of 1665 elite athletes in the Netherlands, Belgium (Flanders), and Germany had experienced a form of sexual violence in organized sports; 10% even reported a severe form of sexual violence (Ohlert et al. 2020). In an Australian study, 41% of female and 29% of male athletes reported experiences of sexual violence within the sports context (Leahy et al. 2002). In

most of the cases, a coach or a fellow athlete was the perpetrator (Bjørnseth and Szabo 2018; Vertommen et al. 2017). Experiences with sexual violence are generally associated with mental health outcomes such as depression, panic disorders, dysfunctional sexual behavior, and post-traumatic stress disorders (e.g., Maniglio 2009). Long-term consequences that have been found in victimized athletes include higher levels of psychological distress and a lower quality of life (Vertommen et al. 2018; Ohlert et al. 2019).

The sports context in Germany differs from that found in other countries, such as the USA, because sport in Germany is largely organized independently of schools and universities. Instead of being affiliated to the educational context, sport is organized in various federations and clubs. Regardless of this, the sports context comprises environmental and social factors that might facilitate the emergence of sexual violence. These factors include, for example, "no pain, no gain" culture, asymmetrical power relationships between coaches and athletes, and the male-dominated gender ratio (Brackenridge 2001). Because of these risk factors and the high incidence of sexual violence, sports clubs and federations have begun to implement preventive measures. In Germany, most of these consist of rules and regulations, such as recommendations on dealing with cases of sexual violence in sports clubs and a code of conduct for coaches. Other often-recommended preventive measures include establishing a contact person for child protection or holding workshops for coaches and athletes (Deutsche Sportjugend [dsj] im Deutschen Olympischen Sportbund e.V. 2013). However, in order to implement these measures in the clubs, it is first necessary to sensitize the club management and coaches to the topic of sexual violence. Thus, the German Youth Sport Organization (dsj) has created a workshop concept for this purpose (Deutsche Sportjugend [dsj] im Deutschen Olympischen Sportbund e.V. 2015).

The crucial objective for sensitizing interventions and workshops is for the measures to actually be effective in changing the behavior of the workshop participants; specifically that they will actively engage in strategies against sexual violence in their organizations. However, there is a lack of research on preventive interventions and measures against sexual violence within the sports context (Ohlert et al. 2019). Most studies on the effectiveness of interventions against sexual violence in sports deal with sexual violence in dating among intercollegiate or high school athletes in North America (Jaime et al. 2015; Miller et al. 2012; Moynihan et al. 2010). These studies show that intervention programs on sexual violence in dating can significantly contribute to increasing athletes' confidence and intention to act toward ending sexual violence in dating and to creating more positive bystander behavior (e.g., how athletes responded to witnessed behavior among peers or friends; Jaime et al. 2015; Miller et al. 2012; Moynihan et al. 2010). A systematic review of DeGue et al. (2014) found that most of the existing preventive interventions against sexual violence were implemented within the educational context (i.e., in schools or universities). Furthermore, the authors reported that the effectiveness of the interventions was only tested in three of the included studies. The authors conclude that future research should evaluate interventions based on an appropriate theoretical framework.

The theory of planned behavior (TPB; Ajzen 1991) has been established as a theoretical framework for explaining and predicting behavior and especially behavior change (Steinmetz et al. 2016); thus, it can serve as a framework to evaluate the effectiveness of interventions against sexual violence. According to the TPB, an individual's intention is related to the performance of a certain behavior, and is described as an antecedent of behavior (Ajzen 1991). An intention is built through the influence of the following motivational variables and key elements of TPB: the attitude toward a behavior, the perceived subjective norms in the actor's peer group, and the subjectively perceived behavioral control (Ajzen 2012; Steinmetz et al. 2016). The attitude toward a behavior consists of an individual's beliefs, evaluations, and behavioral inclinations (i.e., what do I think about the prevention of sexual violence in sport clubs?). Subjective norms or social pressure are beliefs about others' expectations (i.e., what do the others think about the prevention of sexual violence in sport clubs?). Perceived behavioral control is defined as the amount of control an individual believes that she or he has in performing a certain behavior (Ajzen 2012, i.e., how

do I assess my own possibilities to influence the prevention of sexual violence in sport clubs?). According to Bosnjak et al. (2020), the following relationship can be assumed between the key elements of attitude, subjective norms, and perceived behavioral control and intention: "...the more favorable the attitude and subjective norm, and the greater the perceived control, the stronger should be the person's intention to perform the behavior in question" (p. 353). With regard to behavior change and the prevention of sexual violence, this means that interventions should aim at addressing a more positive attitude and social norm as well as enhancing the perceived control of an individual to build a strong intention. Using TPB in the field of the prevention of sexual violence in sport and to evaluate the effectiveness of an intervention helps to gain insights into which key elements of behavior change have been influenced by the intervention. With this insight it might also be possible to identify which key elements need to be addressed further by the intervention to enhance the likelihood of behavior change.

TPB has been applied to predict behavior and behavior change in different health-related behaviors (e.g., exercise behavior, alcohol consumption, nutrition-related behaviors) and has been examined in different contexts (e.g., economics: Yadav and Pathak 2017; physical activity: Prapavessis et al. 2015; education: de Leeuw et al. 2015). Several systematic reviews and meta-analyses that summarize intervention studies based on TPB and their effects are available in the literature (e.g., Tyson et al. 2014; Hackman and Knowlden 2014; Webb et al. 2010; Hardeman et al. 2002). A combined systematic review and meta-analysis by Webb et al. (2010) analyzed intervention studies on using the internet to promote changes in health behavior (physical activity, dietary behavior, alcohol consumption, and smoking abstinence). They found that intervention studies that used TPB as a theoretical framework yielded larger effects on behavior than studies that were grounded in other theories. In the meta-analysis of Steinmetz et al. (2016), 123 interventions were analyzed that were based on TPB and were conducted in the behavioral domains of physical activity, nutrition, work or school, alcohol and drug use, adherence to medical regimens, driving, sexual behavior, and hygiene. The study revealed that the interventions had significant effects, with effect sizes varying from small to medium in most domains.

In several studies, TPB was expanded to include additional factors that gave insights into behavior change mechanisms. For example, in cross-sectional studies on TPB and drinking behavior in athletes and students, social-situational factors such as peer influence, drinking culture among peer groups, the presence of a person of authority (e.g., the coach), group cohesion (Jamison and Myers 2008; Ohlert and Kleinert 2014), and social support (Stapleton and Martin Ginis 2014), were added. Furthermore, in an intervention study on the promotion of physical activity and healthy eating in older adults with diabetes and cardiovascular diseases (White et al. 2012), planning was added as an additional factor of TPB. Planning is seen as a volitional variable that specifies where, when, and how a behavior is to be performed (White et al. 2012).

For the present study, it was helpful to add some factors with a direct influence on TPB in order to increase the validity of the model. Thus, the participants' own experiences with sexual violence, their sports clubs' culture of prevention, and their own knowledge about sexual violence were added as additional factors of TPB. The participants' own experiences with sexual violence were additionally assessed, because personal experiences are part of the development of attitudes. The sports clubs' methods of prevention were measured because they might influence the formation of subjective norms. Knowledge about sexual violence was measured because comprehensive knowledge might influence the estimation of perceived behavioral control.

To sum up, intervention studies on preventing sexual violence are very limited, particularly within the sports context. Furthermore, the effectiveness of existing interventions has not been tested in most cases. To fill these research gaps, the purpose of the present study was to test the short-term and long-term effects of an existing workshop for sensitizing actors in sports to sexual violence with the help of TPB. It should be tested in greater detail how the TPB-related factors (knowledge, attitudes, perceived behavior control, intention)

changed before and immediately after a workshop, and whether those changes might be stable after six months. Additionally, it was of interest whether the workshop might contribute to more protective measures against sexual violence in the participants' sports clubs and a to a positive change of the prevention culture. Based on this, we assumed the following hypotheses:

I.      The workshops will have a positive short-term influence on TPB factor attitudes towards sexual violence, perceived behavioral control and intention, and the additionally measured factor of knowledge about sexual violence.

II.     Short-term changes in these factors will be stable after six months.

III.    Six months after the workshops, more protective measures against sexual violence will be implemented in the club and by the participants themselves, and subjective norms as well as the prevention culture of the club will have changed positively.

IV.    Correlations between different TPB constructs as proposed by TPB will be detected.

## 2. Materials and Methods

### 2.1. Participants

The participants included 137 stakeholders in sports: 66% coaches, 41% athletes, 24% board members, 13% supervisors of the team, 10% parents, 5% contact persons for child protection, and 8% with another relationship to the sports organization (multiple answers were possible). The participants were 35.3 years old on average ($SD$ = 14.7, min = 14, max = 75). Male persons were overrepresented at 69%, compared to female participants at 31%. Participants indicated before the workshop that they had little prior knowledge of sexualized violence ($M$ = 2.67, $SD$ = 1.64, the scale ranged from 1 = very low to 7 = very high). All participants took part in one of seven workshops against sexual violence and filled out a questionnaire before and directly after the workshop. Of these participants, 40 also responded to the online questionnaire six months after the workshop. They had a mean age of 39.2 years ($SD$ = 15.2); 74% were male and 26% were female. The majority of them (59%) were coaches, followed by athletes (54%), board members (33%), supervisors (23%), parents (13%), contact persons for child protection (8%), and others (10%).

### 2.2. Measurements

The measurements were created according to the prescriptions on constructing TPB questionnaires by Ajzen (2006). The factors of "prevention culture in sports club" and "knowledge about sexual violence in sports" has been added, because it can be assumed that the prevention culture can have an effect of the development of the subjective norm. Knowledge about sexual violence in sports is assumed to be a factor that might have an effect on the attitudes toward sexual violence. According to Ajzen's instruction (2006), other factors can be added to the theory that might have an influence on the intentionto change a certain behavior. Therefore, all measured constructs relate to the TPB.

Attitudes toward sexual violence in sports. Participants' attitudes toward sexual violence in sports were assessed by four items. The steamof these items was "I believe that measures on sexual violence are . . . " Participants had to rate each item on a seven-point scale, from 1 = useless to 7 = useful; 1 = necessary to 7 = unnecessary; 1 = important to 7 = unimportant; and 1 = bad to 7 = good. The reliability for this factor was $\alpha$ = 0.83 for the current sample.

Subjective norm. With the help of one item, participants were asked to rate the amount of pressure they felt from other people to implement measures against sexual violence. The item had to be rated on a 10-point Likert scale from 1 ("very low pressure") to 10 ("very high pressure").

Perceived behavioral control. Participants' perceived behavioral control in implementing preventive measures in their sports club or organization was measured by four items. A sample item was "It is up to me to implement a prevention measure on sexual violence in my sports club or association." A seven-point Likert scale from 1 ("does not apply at

all") to 7 ("applies totally") was used for this item. The reliability for this factor with the current sample was $\alpha = 0.72$.

Intention. Three items were used to measure participants' intention to implement measures against sexual violence. The participants were asked to rate whether they intend, try, and plan to implement measures against sexual violence within the following six months. The Likert scale ranged from 1 ("does not apply at all") to 7 ("applies totally"); the reliability for this factor was $\alpha = 0.92$.

Behavior. To measure the actual behavior regarding the prevention of sexual violence in sports, participants were asked how many measures to prevent sexual violence they had personally implemented in any sports club or association within the last six months (their own behavior), and how many preventive measures had been installed in their club (independent of themselves; club behavior).

Prevention culture in the sports club. The prevention culture in participants' sports clubs was assessed by asking whether there was a contact person for complaints of sexual violence, whether there were defined rules for coaches for their behavior with athletes, and whether there were defined rules in the case of a suspicion of sexually violent behavior in the club. The seven-point Likert scale for these questions ranged from 1 ("does not apply at all") to 7 ("applies totally"). The Cronbach's alpha for these three questions was acceptable: 0.75.

Knowledge about sexual violence in sports. To measure participants' knowledge of sexual violence in sports, 15 multiple-choice questions on factual knowledge (e.g., characteristics of sexual violence in sports) were developed by a group of experts on sexual violence in sports. The item difficulty was assessed by a pre-test, and the six most difficult items were selected for the main study. Each item consisted of two statements. The participants were asked to decide whether only one statement was true, both statements were true, or neither of the statements were true. The number of correct answers was totaled to yield a value between 0 and 6.

### 2.3. Intervention Concept

The tested intervention concept against sexual violence in sport was developed by the German Sport Youth Organization (dsj), which is responsible for the prevention of sexual violence in organized sports in Germany. The intervention concept was not developed based on TPB. The TPB was used as a theoretical background to examine and explain the effects on behavioral change in this study. The concept was developed for experts in the area of sexual violence in order to provide them with valid material for workshops against sexual violence in sports clubs The intervention concept consists of two parts. The first part of the intervention aims at raising participants' awareness of situations of sexual violence in sports. During this part of the intervention, the participants learned how to define and perceive situations of sexual violence, the German legal basis for defining sexual violence, and the characteristics and strategies of perpetrators. During the second part of the intervention, the participants learned key elements of the prevention concept within a sports club. Subsequently, the participants applied this knowledge in small groups and developed practical steps for implementing a prevention measure within their sports club or association. The last part of the intervention dealt with the handling of suspected cases with the help of counseling centers. A detailed description is available for download on the web page of the dsj (the link can be found in the Supplementary Materials section).

### 2.4. Procedure and Design

The study followed the ethical guidelines of the American Psychological Association. A request for support for the study was posted on the webpage of the dsj where the intervention concept was listed. Furthermore, the dsj sent emails to their known lecturers on sexual violence and asked them for their help. In total, nine different lecturers agreed to support the study by giving the questionnaires to their participants. Participants were vol-

unteers of different sport organizations who had voluntarily registered for the workshops of the different lecturers. The workshops were free to the participants.

The lecturers were asked to give questionnaires to the participants in their workshops directly before (t1) and after (t2) their intervention workshops, and to encourage them to participate. With the questionnaires, an informed consent form was distributed that participants had to sign before they could start filling out the questionnaires. On the consent sheet, they could give their email address in case they wanted to participate in the third part of the study. They were informed that participation was voluntary and that they could abandon the survey at any time without consequences. No one received any compensation for participation in t1 and t2 of the study. Consent sheets and questionnaires were collected separately by the lecturers and sent to the authors. All participants who had given their email address received an email six months after the intervention workshop, in which they were asked to participate in an online questionnaire regarding the intervention (t3). People participating at t3 could enter a draw to win a prize of EUR 100 after completing the questionnaire. Again, informed consent had to be given on the first page of the online questionnaire. At t1 and t3, all constructs (attitude, subjective norm, perceived behavioral control, intention, behavior, prevention, culture, and knowledge) were included in the questionnaire. At t2, the questionnaire only included attitude, perceived behavioral control, intention, and knowledge about sexual violence, because the other constructs (i.e., changes in prevention culture, subjective norms and behavior) could not have been affected immediately after the workshop.

*2.5. Data Analysis*

The data were analyzed using the IBM software SPSS Statistics 25.0. Prior to the analyses, a check for duplicates, implausible answers, and multivariate outliers according to Tabachnick and Fidell (2013) was performed. Factors for each point of measurement and all scales were built by calculating the mean values. The dropout analysis was performed by comparing the participants who dropped out of the sample with those who were retained via *t*-tests on age, knowledge at t1 and t2, intention at t1 and t2, and attitude at t1 and t2. Due to the small number of participants at t3, it was decided to undertake the analyses for the short-term effects separately from those with regard to the long-term effects.

To test if the factors of attitude, behavioral control, intention, and knowledge changed pre (t1) or immediately after the workshop (t2), and after six months (t3), a multivariate Analyses of Variance (MANOVA) was run. For the short-term effects, a MANOVA was performed with attitude, behavioral control, intention, and knowledge as dependent variables, and time (before (t1) and after the workshop (t2)) as the independent variable. For the long-term effects of the same factors, another MANOVA was computed with the same dependent variables, but all three points of measurement were included (t1, t2, and t3). T-tests for dependent samples were used as post-hoc tests to clarify which points of measurement produced the respective effects. To analyze the differences between the constructs measured only at t1 and t3 (subjective norm, participants' own behavior, and culture of prevention), a third MANOVA was calculated with the respective constructs as dependent variables, and the time between t1 and t3 as the independent variable. For club behavior, only 23 persons were able to tell how many measures against sexual violence the club had taken at t3; therefore, this variable was analyzed separately, with a *t*-test for the dependent variables.

In order to analyze the TPB-predicted interrelations between the different constructs, structural equation modeling was planned. However, as the number of participants was too small to analyze the whole model, a regression model was calculated with intention at t2 as a criterion, and attitude (t2), subjective norm (t1), and perceived behavioral control (t2) as predictors. A second regression was planned with behavior at t3 as the criterion, and intention (t2) and perceived behavioral control (t2) as predictors. Again, due to the limited number of cases, this was not possible, particularly for club behavior, and Pearson correlations were computed instead.

## 3. Results

### 3.1. Dropout Analysis and Descriptive Measures

The dropout analysis revealed that the persons who did not participate at t3 were significantly younger than the participants retained in the sample ($t_{(133)} = -2.04$, $p = 0.043$; see Table 1). For the remainder of the tested variables, no differences could be found between the two groups.

**Table 1.** Results of the dropout analysis (participants who dropped out between t2 and t3).

| | Retaining (*n* = 40) M (SD) | Dropout (*n* = 97) M (SD) | $t_{(133)}$ | *p* |
|---|---|---|---|---|
| Age | 39.2 (15.2) | 33.6 (14.3) | −2.04 | 0.043 |
| Attitude t1 | 5.95 (1.15) | 5.89 (1.31) | −0.25 | 0.800 |
| Attitude t2 | 6.31 (0.95) | 6.14 (1.36) | −0.72 | 0.474 |
| Intention t1 | 4.03 (1.77) | 3.76 (1.79) | −0.80 | 0.424 |
| Intention t2 | 5.16 (1.62) | 4.57 (1.73) | −1.86 | 0.067 |
| Knowledge t1 | 2.45 (1.28) | 2.44 (1.22) | −0.03 | 0.977 |
| Knowledge t2 | 3.34 (0.92) | 3.25 (1.39) | 0.38 | 0.707 |

Means and standard deviations for all relevant variables and all three points of measurement are depicted in Table 2.

**Table 2.** Means and standard deviations for all constructs measured at all points of measurement for the whole sample.

| | t1 (*n* = 137) M (SD) | t2 (*n* = 137) M (SD) | t3 (*n* = 40) M (SD) |
|---|---|---|---|
| Attitude | 5.91 (1.26) | 6.29 (1.25) | 5.95 (1.33) |
| Subjective norm | 3.85 (2.53) | - | 3.61 (2.26) |
| Perceived behavioral control | 4.99 (1.22) | 5.58 (1.17) | 5.29 (1.23) |
| Intention | 3.84 (1.78) | 4.74 (1.71) | 3.61 (1.96) |
| Own behavior | 0.09 (0.32) | - | 0.58 (1.14) |
| Club behavior | 0.45 (0.92) | - | 1.35 (1.61) |
| Knowledge | 2.45 (1.23) | 3.31 (1.08) | 3.17 (1.22) |
| Culture of prevention | 4.70 (1.75) | - | 5.61 (1.54) |

### 3.2. Short-Term Effects of the Intervention (t1, t2)

Looking at all dependent variables together, i.e., on a multivariate level, the MANOVA found a significant main effect of the intervention (factor time: $F_{(4,131)} = 33.14$, $p < 0.001$, $\eta^2_{(part)} = 0.50$) with a large effect size. This large effect size indicates a solid overall effect of the workshop between t1 and t2. At the level of each individual dependent variable, i.e., at the univariate level, the main effect could be found on all four measured constructs, with a small effect size for the attitude ($F_{(1,134)} = 9.58$, $p = 0.002$, $\eta^2_{(part)} = 0.07$), but a large effect size for perceived behavioral control ($F_{(1,134)} = 43.54$, $p < 0.001$, $\eta^2_{(part)} = 0.25$), the intention to act against sexual violence ($F_{(1,134)} = 55.73$, $p < 0.001$, $\eta^2_{(part)} = 0.28$), and knowledge about sexual violence ($F_{(1,134)} = 50.70$, $p < 0.001$, $\eta^2_{(part)} = 0.29$).

### 3.3. Long-Term Effects on Attitude, Perceived Behavioral Control, Intention, and Knowledge (t1, t2, t3)

On the multivariate level, the MANOVA revealed a significant overall main effect of time, with a large effect size ($F_{(8,29)} = 6.39$, $p < 0.001$, $\eta^2_{(part)} = 0.64$). On the univariate level,

the effect was repeated for all four measured constructs with three points of measurements: a medium-sized effect was found for the attitude ($F_{(2,72)} = 3.35$, $p = 0.047$, $\eta^2_{(part)} = 0.09$); for the other constructs, however, the effect size was large (perceived behavioral control: $F_{(2,72)} = 7.24$, $p = 0.001$, $\eta^2_{(part)} = 0.17$; intention: $F_{(2,72)} = 17.00$, $p < 0.001$, $\eta^2_{(part)} = 0.32$; knowledge: $F_{(2,72)} = 5.77$, $p = 0.005$, $\eta^2_{(part)} = 0.14$).

Table 3 shows the statistical values of the calculated post hoc tests in order to identify the location of the significant effects of time. In addition to the fact that all constructs changed significantly from t1 to t2 for this smaller sample as well (cf. the results above), a significant difference between t1 and t3 (long-term effects) could only be detected for the factor knowledge about sexual violence (medium effect size). For the three TPB constructs, the effect was not retained. For perceived behavioral control and intention, there was a significant decrease between t2 and t3 (medium to large effect size).

**Table 3.** Statistical values of the post-hoc *t*-tests for the subsample with all three points of measurement.

| | t1 vs. t2 | | | t1 vs. t3 | | | t2 vs. t3 | | |
|---|---|---|---|---|---|---|---|---|---|
| | $t_{(36)}$ | $p$ | $d$ | $t_{(36)}$ | $p$ | $d$ | $t_{(36)}$ | $p$ | $d$ |
| Attitude | −3.16 | 0.003 | 0.36 | −0.32 | 0.749 | | 1.91 | 0.064 | 0.28 |
| Perceived Behavioral Control | −3.88 | <0.001 | 0.72 | −0.69 | 0.497 | | 2.80 | 0.008 | 0.56 |
| Intention | −5.16 | <0.001 | 0.65 | 1.38 | 0.178 | | 4.79 | <0.001 | 0.84 |
| Knowledge | −3.73 | 0.001 | 0.61 | −2.95 | 0.005 | 0.57 | 0.41 | 0.682 | |

Note: effect sizes are only shown for $p < 0.1$.

### 3.4. Long-Term Effects on Subjective Norm, Own Behavior, Club Behavior, and Culture of Prevention (t1, t3)

When focusing on the long-term effects between t1 and t3 regarding subjective norms, participants' own behavior, and culture of prevention in the club, the MANOVA demonstrated a significant overall main effect of time (i.e., a long-term effect of the intervention) on the multivariate level, with a large effect size ($F_{(3,30)} = 6.39$, $p = 0.002$, $\eta^2_{(part)} = 0.38$). On the univariate level, however, the effect could only be found for the culture of prevention, with a large effect size ($F_{(1,32)} = 11.81$, $p = 0.002$, $\eta^2_{(part)} = 0.27$). For participants' own behavior, the effect did not reach significance due to the small sample size ($F_{(1,32)} = 4.11$, $p = 0.051$, $\eta^2_{(part)} = 0.11$). The subjective norm did not change over half a year ($F_{(1,32)} = 11.81$, $p = 0.002$, $\eta^2_{(part)} = 0.27$).

For club behavior, the *t*-test for dependent samples produced a significant effect with a large effect size ($t_{(17)} = -2.58$, $p = 0.019$, $d = 0.84$), indicating that more measures against sexual prevention had been introduced in the clubs within the six months after the intervention.

### 3.5. Interrelations between the Constructs According to TPB

The linear regression analysis with intention at t2 as the criterion revealed a large effect size of the overall model ($F_{(3,131)} = 23.83$, $p < 0.001$, $R^2 = 0.34$). All three predictors reached significance (see Figure 1). The second regression with participants' own behavior at t3 as a criterion did not reach significance ($F_{(2,33)} = 2.28$, $p = 0.118$). However, the simple correlations between the intention and perceived behavioral control at t2, on one hand, and participants' own and club behavior at t3, on the other hand, revealed that participants' own behavior at t3 correlated with both TPB constructs (intention: $r = 0.33$, $p = 0.047$; perceived behavioral control: $r = 0.22$, $p = 0.191$). The same held true for club behavior (intention: $r = 0.37$, $p = 0.086$; perceived behavioral control: $r = 0.41$, $p = 0.049$) with all correlations of a medium effect size.

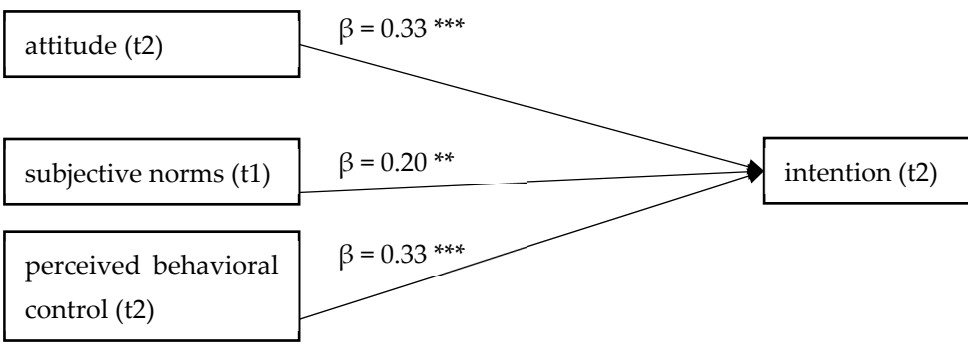

**Figure 1.** Results of a regression analysis on the interrelations of TPB-constructs with intention (t2) as the criterion; *** $p < 0.001$, ** $p < 0.01$.

## 4. Discussion

The aim of the present study was to test the short- and long-term effects of a workshop intervention on the prevention of sexual violence in the context of organized sports in Germany. It was hypothesized that the workshop intervention would have positive short-term effects on the TPB-related factors (knowledge, attitudes, perceived behavior control, and intention), and that those effects would be stable after six months. Moreover, it was hypothesized that the workshop intervention would lead to the implementation of more protective measures against sexual violence by the participants themselves in their respective sports clubs, and that the subjective norms and prevention culture would change positively. With regard to short-term effects, the results showed that the workshop intervention had a positive effect on the four TPB factors, as was hypothesized. With regard to long-term effects, it was found that only the factor of knowledge about sexual violence improved over six months. Moreover, it was determined that the culture of prevention changed positively, and that more measures against sexual violence were implemented in the sports clubs in the long term.

As predicted by hypothesis I, our analyses showed that the workshop intervention increased the factors of attitude, perceived behavioral control, intention to act against sexual violence, and knowledge about sexual violence in the short term. These results indicate that the workshop can help to sensitize people to the topic of the prevention of sexual violence in sports in the short term, and are an important basis for long-term changes. The relevance of these results is underlined by the large effect sizes (Cohen 1988).

Hypothesis II can only be partially confirmed, because the stable long-term effects of the workshop interventions could be detected solely for the factor of knowledge about sexual violence; while attitude, perceived behavioral control, and intention to act against sexual violence decreased again in comparison with measures taken directly after the workshop. Several explanations are possible. First, when looking specifically at the factor of perceived behavioral control, it is possible that the participants tried to implement preventive measures in their respective sports club, but encountered resistance and concluded that they did not have as much control as they had assumed, which would lead to decreasing perceptions of behavioral control. With regard to the factor of intention to act against sexual violence, a possible explanation is that the participants had successfully implemented preventive measures in their sports clubs (as can be demonstrated within our analyses), and thus their intention to implement even more preventive measures was no longer as high after six months. An alternative explanation is that having just one session of a workshop intervention is not sufficient to effectively change stable factors such as personal attitude, perceived behavioral control, and intention to act against sexual violence in the long term. However, the fact that more measures against sexual violence were taken in the clubs speaks in favor of the first explanation.

With regard to hypothesis III, the analyses of long-term changes of the factors of subjective norms, the culture of prevention, and behavior (participants' own behavior and club behavior), a more positive culture of prevention and more positive club behavior

six months after the intervention were recorded. The changes toward a more positive prevention culture and more positive club behavior are very important results, because they indicate that the intervention did indeed contribute to a behavior change. However, with regard to club behavior, it should be kept in mind that there might have been other reasons that led to an increase in preventive measures as well, such as external pressure (e.g., by media or politics) or cases of sexual violence within a particular sports club. As we did not conduct an experimental study, causal interpretations should be handled with care.

The subjective norms did not change over time, and neither did the number of protective measures implemented by the participants themselves. The latter result might be caused by the small sample size, as the effect was of a medium size and almost reached significance. The reasons for no long-term changes in the subjective norms might lie in the fact that this construct was not targeted within the workshop intervention and that it is highly dependent on the behaviors and attitudes of a person's peers (Ajzen 1991). Thus, it was not unexpected that the subjective norms the participants perceived within their surroundings did not change significantly over time.

For the design and research of further interventions, our results on the interrelations between the different TPB constructs can be an important cue. Even though the sample size was too small to compute structural equational models, we believe that our study was the first to apply TPB to a workshop intervention in order to change behaviors in the area of sexual violence, and thus provides important conclusions. As could be derived from our results, TPB might provide a helpful framework for designing interventions regarding long-term behavior change in sexual violence for sports organizations. However, it should be mentioned that TPB refers solely to behavioral changes at the individual level. Changes at the contextual level are not taken into account. However, these are important for the prevention of sexualized violence in sport, and should be considered. To address this, changes in the prevention culture were assessed in the present study, although not in detail. Further studies with larger sample sizes are necessary to support our results.

*4.1. Strengths and Limitations*

The results of the present study showed that a workshop intervention can have positive effects on the behavior of different stakeholders within organized sports with regard to the prevention of sexual violence. Two major strengths of the study can be mentioned: First, the study contributes to filling the gap in intervention studies on the prevention of sexual violence in the sports context and offers insights into the effectiveness of such interventions in sports clubs. Second, our study was the first to prove that TPB can serve as an effective theoretical background for interventions against sexual violence.

Besides these strengths of the present study, a few limitations must be identified. First, the methodological design of the study only included an intervention group and did not use a control group. Therefore, alternative explanations for the changes found in our study might be possible. One explanation might be the fact that the sports organizations received pressure from the public to initiate interventions against sexual violence. However, as our effects can also be demonstrated for those factors influencing behavior change, we still believe that our results give an important hint for further research and interventions. Another explanation might be that the workshop served those people who were already committed to the need for the prevention of sexual violence in their sports club. With regard to the methodological design of the study, it should be mentioned again that the workshop concept was not developed based on TPB. The TPB was only used to evaluate and explain behavioral effects. Second, the sample size at t3 remained very small, so we were not able to perform all analyses as planned. However, it has to be noted that it is seldom the case that the same workshop is held several times with a homogenous sample; thus, our study still comprises an exceptional sample. Furthermore, except for age, no differences in important variables could be found for the persons who dropped out of the study.

As another limitation, it has to be noted that the evaluated workshop conception had not been developed based on the TPB. Thus, an intentional matching of workshop parts to the different factors of the TPB was not possible. However, it has to be noted that even though there was no theoretical matching, the workshops still fostered the different TPB factors necessary for behavior change. Thus, it can be argued that a workshop which would have been built explicitly based on the TPB should produce even stronger effects.

### 4.2. Implications for Research and Practice

Our study provides new insights into behavior change processes concerning the prevention of sexual violence in sports organizations. However, as the results indicate, future studies—with a larger sample size—are needed to investigate whether the current findings can be replicated. Furthermore, how interventions could improve changes in stable constructs such as attitude or subjective norms should be investigated. In particular, it might be necessary to have more than one workshop session over a longer period of time. Moreover, randomized controlled studies are necessary to shed light on the causal relationships between the contents of the intervention and the different factors indicating behavior change.

Concerning the practical implications of the results, this was one of the first studies to prove the effectiveness of an intervention against sexual violence in sports. Thus, it can be stated that a single workshop session is helpful in improving the stakeholders' prevention behavior with respect to sexual violence in their respective sports clubs, and even in producing medium-sized effects. This is good news for those stakeholders in sports addressing sexual violence, as relatively few efforts have been shown to have positive effects and to increase the importance of the topic within the clubs. Furthermore, future interventions could be designed with an even sharper focus on targeting TPB constructs, such as changing attitudes or increasing the perceived behavioral control with regard to measures against sexual violence. Thus, our study can be seen as one step further on the path to eliminating sexual violence in sports.

**Supplementary Materials:** A detailed description of the intervention concept "dsj-Qualifizierungsmodul 'Gegen sexualisierte Gewalt im Sport'" and additional material (in German) can be download at: https://www.dsj.de/themen/kinder-und-jugendschutz/qualifizierungsangebote (accessed on 12 April 2023).

**Author Contributions:** Conceptualization, M.A. and T.R.; methodology, J.O.; formal analysis, J.O.; writing—original draft preparation, A.S.-P. and J.O.; writing—review and editing A.S.-P., J.O., M.A. and T.R.; project administration, J.O., M.A. and T.R.; funding acquisition, M.A. All authors have read and agreed to the published version of the manuscript.

**Funding:** This research was funded by the German Federal Ministry of Education and Research under Grant number 01SR 1401XY.

**Institutional Review Board Statement:** The study was conducted in accordance with the Declaration of Helsinki, and approved by the Ethics Committee of the University of Ulm.

**Informed Consent Statement:** Informed consent was obtained from all subjects involved in the study.

**Data Availability Statement:** The data presented in this study are available upon request from the corresponding author. The data are not publicly available due to ethical considerations with respect to sensible data.

**Conflicts of Interest:** The authors declare that they have no conflict of interest. The funders had no role in the design of the study; in the collection, analysis, or interpretation of data; in the writing of the manuscript; or in the decision to publish the results.

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
