# Peer review of "Short- and Long-Term Effects of an Intervention to Act against Sexual Violence in Sports"

_socsci, doi:10.3390/socsci12040244_

Round 1

Reviewer 2 Report

Comments to the Author

The manuscript is well intended, interesting and well written. It addresses an important and under researched topic. However, in its current state it requires a bit more work to be publishable in Social Sciences and to make it a genuine contribution to the field. There are several reasons for the above conclusion.

1. Introduction

The introduction is fine, but the broader context needs more elaboration. For instance, it would be nice to know about other efforts/measures to reduce sexual abuse in sport in Germany. Some information about participation in sport for girls and boys would also be nice.

The actual workshop concepts also need to be presented in the introduction. Hence, point 2.3 Intervention concept could be part of the introduction. We also need more information about the actual activities – and the wanted outcomes – of the workshops.

2. Theory

The selected theory in the paper is The theory of planned behavior – TPB- and it is justified, with reference to TPB being widely used, and especially relevant when studying behavioral change. The paragraph describing TPB (79-90) should be developed. Some sentences are obvious e.g. “an individuals intention is related to the performance of a certain behavior and is described as an antecedent of behavior” (82-83). We need to learn more about the specific contribution from TPB.  Since behavioral change is the topic of the paper – how this is captured in TPB should be explicitly explained. 

Other questions related to the theory:

-          Were the workshops developed based on TPB?

-          The limitations of TPB should be mentioned: As presented in the article TPB seems to cover contextual factors only to a limited degree. This should be discussed – especially since certain aspects of sport culture is relevant for harassment and abuse in sport, e.g. probably makes it harder to report sexual harassment and abuse.

-          The discussion pivots around four TPB-related factors: perceived behavior control, intention to act, knowledge, attitude. Are these key factors in TPB? If so – this should be explained in the theory section.

3. Methods.

The study design seems appropriate. The intervention was evaluated by measuring the short-term (immediately before and after the workshop) and long-term effects (after six months) through questionnaires.

Some topics are lacking in the method section.

-          We need to know how the participants were recruited: How do they possibly differ from the general population? What is their class background etc. Are the ones participating in the workshop – and the study - particularly interested in the field of sexual harassment in sport?

-          The actual questions addressed to the participant are not sufficiently described. A table including the actual questions would be welcomed.

-          Were the same measures used at t1, t2 and t3?

-          How do the seven listed measures (p. ) relate to TPB?

-          The relevance of the actual questions to measure the wanted outcomes of the workshop should be discussed: How do the seven listed measures relate to the wanted outcomes of the workshops?

Due to limited knowledge about the statistical analyses, I cannot give much feedback on them. Still, I will suggest that the presentation of the results could be more adjusted to people without statistical knowledge. E.g., point 3.1 – you could give us the concrete numbers – that would make sense to all readers.

4. Findings and discussion.

Some of the comments to the methodology could be addressed in the discussion paragraphs.

The presentations of the results could be better adjusted to readers with less skills in statistical analyses.

Some minor comments:

-          Page 6, line 278 “univariate level” – is this correct? Isn’t this a bivariate level – revealing the statistical relationship between participating in the workshop and outcome measures.

-          Page 12, line 370: references are needed

-          The sentence (line 156-158) does not make sense to me. You need to explain which of the measures that relates to TPB

This article really has potential. I hope my comments are considered valuable for the improvement of the article. This will hopefully be a contribution to the literature on safeguarding in sport.
